# Docking and Molecular Dynamics Predictions of Pesticide Binding to the Calyx of Bovine β-Lactoglobulin

**DOI:** 10.3390/ijms21061988

**Published:** 2020-03-14

**Authors:** Paulina Cortes-Hernandez, Roberto Vázquez Nuñez, Lenin Domínguez-Ramírez

**Affiliations:** 1Instituto Mexicano del Seguro Social (IMSS), Centro de Investigación Biomédica de Oriente (CIBIOR), Cellular Biology Laboratory, 74360 Metepec, Puebla, Mexico; paulina.cortes.hernandez@gmail.com; 2Department of Fundamental Microbiology, University of Lausanne, CH-1015 Lausanne, Switzerland; RobertoJareth.VazquezNunez@unil.ch; 3Department of Chemical and Biological Sciences, School of Sciences, Universidad de las Américas Puebla, Santa Catarina Mártir Cholula, 72810 San Andrés Cholula, Puebla, Mexico

**Keywords:** docking, molecular dynamics, MMGBSA, cypermethrin, pyrethroid pesticide

## Abstract

Pesticides are used extensively in agriculture, and their residues in food must be monitored to prevent toxicity. The most abundant protein in cow’s milk, β-lactoglobulin (BLG), shows high affinity for diverse hydrophobic ligands in its central binding pocket, called the calyx. Several of the most frequently used pesticides are hydrophobic. To predict if BLG may be an unintended carrier for pesticides, we tested its ability to bind 555 pesticides and their isomers, for a total of 889 compounds, in a rigid docking screen. We focused on the analysis of 60 unique molecules belonging to the five pesticide classes defined by the World Health Organization, that docked into BLG’s calyx with ΔGs ranging from −8.2 to −12 kcal mol^−1^, chosen by statistical criteria. These “potential ligands” were further analyzed using molecular dynamic simulations, and the binding energies were explored with Molecular Mechanics/Generalized Born/Surface Area (MMGBSA). Hydrophobic pyrethroid insecticides, like cypermethrin, were found to bind as deeply and tightly into the calyx as BLG’s natural ligand, palmitate; while polar compounds, like paraquat, were expelled. Our results suggest that BLG could be a carrier for pesticides, in particular for pyrethroid insecticides, allowing for their accumulation in cow’s milk beyond their solubility restrictions. This analysis opens possibilities for pesticide biosensor design based on BLG.

## 1. Introduction

The Food and Agriculture Organization (FAO) of the United Nations defines pesticides as substances aimed at preventing, destroying or controlling pests [1]. This definition groups a wide variety of organic molecules that constitute the active ingredients in insecticides, rodenticides, herbicides, fungicides, fumigants and similar preparations [1] that are abundant worldwide due to their importance in maximizing crop yields and controlling disease vectors. According to the FAO, at least 4.1 million tonnes of pesticides are released into the environment annually [2], most of which are for agricultural use [3]. Asia and the Americas are the regions that use the most pesticides, both in total and by area of cropland; while China, the USA and Brazil are the top users [2,4]. FAO statistics show that worldwide pesticide use increased in the last three decades, almost doubling since 1990, with a deceleration in the last decade [2,5].

These chemicals, that have been selected for toxicity, can easily become pollutants and impact human and environmental health [4]. Thus, the continuous evaluation of potential pesticide exposure routes and bioavailability to humans and other non-target organisms is important [4,5]. The World Health Organization (WHO) classifies pesticides into four classes (Ia, Ib, II and III) according to their median Lethal Dose (LD50), plus a fifth class, class U, that includes compounds with very high LD50s or that are unlikely to present an acute hazard during normal use [6]. Then, the WHO subclassifies pesticides by their main use (for example, as an insecticide, herbicide, rodenticide, fumigant, etc.), and into 15 different chemical types [6]. Herbicides and insecticides are among the most used pesticides, corresponding to 40% and 33% of global use, respectively [3], and they are also the formulations most associated with human health hazards [6].

Cattle may absorb pesticides from feed or from topical ectoparasiticide treatments [4,7]. For instance, pyrethroid insecticides used as ectoparasiticides can be detected at high levels in the raw milk of a cow for over 15 days after a single application [8], while organophosphates have been reported in homogenized and pasteurized milk samples [9]. Hydrophobic pesticides may concentrate in the hydrophobic phases of milk and/or bind to milk proteins. Thus, humans could be exposed to insecticides through dairy products.

The most abundant protein in cow’s milk is β-lactoglobulin (BLG), which corresponds to 50% of whey and 12% of whole milk proteins [10]. It belongs to the lipocalin family of small extracellular proteins that carry hydrophobic ligands [11]. BLG is relevant to the food industry due to its binding spectrum and abundance in cow’s milk; thus, it has been extensively characterized, biochemically and structurally, through X-ray diffraction (XRD), NMR and computational approaches [12,13,14,15]. 

Each BLG monomer consists of 162 residues (18.3 kDa), folded into eight-stranded antiparallel β-sheets that form a hydrophobic binding pocket called the calyx (shown in Figure 1 and Appendix A), flanked by an α-helix [16]. BLG binds to a variety of nutrients like fatty acids, peptides, sugars and some vitamins (Appendix A) in at least two different binding sites, the most prominent of which is the calyx [15,17,18] (the second site is formed in the interphase between two BLGs when they dimerize [15], and will not be explored here). The affinity for the known hydrophobic ligands in the calyx is in the micromolar range [19]. The polarity of the calyx has been explored and described elsewhere [15,20] (illustrated in Appendix A). A “lid” formed by loop EF can close over the empty calyx at an acidic pH, while it opens under alkaline conditions [21,22]. When the calyx is occupied by ligands, the lid has been detected only in the open conformation [23]. Computational studies using docking, as well as molecular dynamics, have been useful in the characterization of BLG’s atomic interactions with ligands [12,15,17,18,19,20], producing data in close accord with those from thermodynamic and crystallographic experiments, and demonstrating that it is possible to predict binding to this protein by computational methods. Here, we used docking and molecular dynamics to predict if pesticides of common use bind to cow’s milk BLG, and to computationally explore whether this protein could act as a pesticide carrier and potentially contribute to human exposure through dairy.

## 2. Results

### 2.1. Pesticide Docking to β-Lactoglobulin

To explore whether cow’s milk BLG can bind pesticides, we carried out a systematic docking screen of all the pesticides listed by the WHO [6], except those that contain metallic ions. These were excluded for two main reasons: first, the handling of metals by docking programs does not include all of the metals found in pesticides; second, metal interactions could lead to the formation of pesticide oligomers. Without experimental confirmation of the latter, the results for pesticides with metallic ions would be uncertain.

Since there is no structural precedent for pesticide binding to BLG (no BLG structures have been crystalized with a pesticide), the first step to set up a systematic docking screen was the selection of a BLG structure to use as a receptor. This step is crucial for accurate predictions, as has been discussed [15,24]. Out of the 72 XRD structures determined for BLG, 37 have ligands and many have gaps. After inspecting all of the structures, we selected the 10 best, based on stereochemical quality and coverage (all residues visible in the electron density map), six with ligands and four empty. Then, we evaluated which of the best structures was able to dock known ligands in the calyx with affinities closest to those calorimetrically determined. Appendix A shows the binding energies calculated from the docking of 10 fatty acids, which are the known natural ligands of BLG, to the 10 BLG structures tested, and compared to the experimentally determined energies. The best receptor was PDB ID 1GXA (Figure 1), a BLG structure crystallized with palmitate [25], which showed binding energies and ligand positions close to those experimentally determined. Thus, 1GXA was chosen to perform the computational dockings to the pesticides reported here.

After computationally removing palmitate and water molecules from the 1GXA structure, we docked four negative control molecules (benzene, phenol, phosphate and acetate) and 555 unique pesticides (28 class IA, 51 class IB, 193 class II, 104 class III and 179 U, as classified by the WHO), and their isomers, for a total of 889 molecules. The docked pesticides are shown in Figure 2 (all) and Appendix A (separated by class and contrasted to the docking results of BLG’s natural ligands).

To study a subset of pesticides likely to behave as BLG ligands, we decided to further analyze as “probable ligands” all of the pesticides that docked with energies below one standard deviation (−1σ = −8.18 kcal/mol) of the mean affinity found overall in the pesticide docking screen (Figure 2). The statistical nature of this threshold renders it less arbitrary as a way to select a subset of pesticides to study further. Additionally, it also corresponds to a higher affinity than that calculated for the BLG natural ligands through docking into 1GXA, except retinol and retinoate (see Appendix A). For comparison, palmitate docked into 1GXA with an energy of −7.77 kcal/mol (white diamond in Figure 2 and Appendix A), while the negative control molecules had the following values: benzene −5.0 kcal/mol (logP 1.6), phenol −5.0 kcal/mol (logP 1.4), phosphate −3.7 kcal/mol (logP −2.3) and acetate −2.7 kcal/mol (logP 0.1) (Figure 2 and Appendix A).

Sixty unique pesticides and their isomers (118 total molecules) from all of the WHO classes displayed binding energies < −1σ of the mean (summarized in Table 1). The logP of these “probable ligands” ranged from 2 to 8.5 (hydrophobic), with a higher density clustered around palmitate’s logP of 6.26. Pyrethroids constituted the most frequent chemical type identified as “probable ligands”, representing 32% of unique molecules, while the other chemical types identified were coumarins (6.7%) and carbamates (3.3%) (Table 1). A group of very hydrophobic class Ia pesticides displayed the highest affinity in the docking screen (red dots in the lower right corner of Figure 2), including several structurally similar coumarin derivatives like bromadiolone isomers.

Structural visualization was carried out for all of the “probable ligands”. Most of them lodged into BLG’s calyx due to their size and hydrophobicity, giving rise to a good shape complementarity with the receptor, similar to that of palmitate, despite having several rings. Two or three representatives from each class were chosen to sample the chemical diversity of the “probable ligands” (Table 1, last two columns) and are shown in Figure 3 in the BLG calyx.

In the docking screen, no hydrophilic pesticides (logP ≤ 0) behaved as “probable ligands” (none had binding affinities below −1σ) (Figure 2). Paraquat, a class II herbicide of abundant use worldwide, showed the highest affinity among the hydrophilic pesticides, at −7.3 kcal mol^−1^: close to, but above, −1σ, and above palmitate’s affinity. Structural visualization showed that paraquat lodged deep in the BLG calyx (Figure 3F); however, the hydrophobic nature of the calyx [15] (Appendix A) suggested that the high affinity results for polar pesticides could represent false positives or unstable binding. We hypothesized that paraquat would leave the calyx if analyzed with molecular dynamics, in contrast to hydrophobic pesticides, and to test that, we included paraquat in the next steps of our analysis, along with the selected “probable ligands”.

### 2.2. Molecular Dynamics of Pesticide-BLG Complexes

The docking experiments described in the previous section have the limitations inherent to receptor rigidness. In order to refine our docking ranking, position and energetic predictions, we decided to sample representative “probable ligands” in complex with BLG, through molecular dynamics. We simulated the 11 pesticides listed in Table 1 and compared them to a simulation of palmitate. We also simulated the herbicide paraquat that did not qualify as a “probable ligand”, but we used it to monitor the behavior of at least one polar compound that showed some affinity in the screen (−7.3 kcal mol^−1^).

Simulations for these complexes were carried out at 300 K, in explicit TIP4PEW solvent, for 100 ns each, and repeated three times, independently. We selected this water model because it allows protein dynamics close to those experimentally measured by NMR, even if it increases the computational overhead. The negative control, acetate, which docked to BLG with a binding energy of −2.7 kcal/mol, was expelled from the calyx in under 5 ns (not shown). Thus, we explored 100 ns of simulation to observe if the protein-ligand interactions predicted by docking were stable. A total of 3.6 microseconds of simulation with pesticides were explored: 100 ns per 12 BLG-pesticide complexes, in triplicate.

To track the movement of ligands relative to the binding site, we employed a simple metric: measuring the distance from the center of mass (CM) of each ligand to the CM of the ring in BLG′s residue F105, located at the middle of the calyx (shown in Figure 1). We will refer to this distance as the CM distance (CMdist). As a reference, the CMdist between the F105 ring and the sidechain of residue E62, located at the calyx entrance (also shown in Figure 1), was 19.66 ± 0.75 Å. To distinguish ligands moving along the length of the calyx from those leaving it altogether, we used the CMdist between F105 and E62 as a threshold. A ligand-F105 CMdist > 19.66 Å was considered to be outside of the binding site. In Figure 4, we show the CMdist over time for the simulations of 1GXA with the representative pesticides. Three trajectories were analyzed per pesticide and contrasted against palmitic acid (in red, as a reference in each panel of Figure 4). Ten out of the 11 pesticides tested remained below the 19.66 Å threshold in all trajectories, suggesting a stable binding inside the calyx. Only coumatetralyl and paraquat exceeded the CMdist threshold. Coumatetralyl docked at the calyx mouth (Figure 3B) and exceeded the threshold in one out of three trajectories (Figure 4D, black trace), suggesting a tendency to leave the calyx. Paraquat exhibited CMdists above the threshold in all trajectories (Figure 4L), suggesting unstable binding, which contrasted with the pesticides deemed as “probable ligands”—for example, pyrethroids (Figure 4C,E,I,J)—which maintained CMdists similar to or lower than that of palmitate.

This information was further analyzed by plotting frequency histograms of the CMdist visited by each ligand (Appendix A), which exhibited the peaks summarized in Table 2. Overall, the analysis suggests that most pesticides identified by docking as having high affinities for BLG (“probable ligands”) remain inside the calyx during the simulated time (a total of 300 ns).

Analyses of BLG’s RMSD (Root Mean Square Deviation), gyration radius or solvent accessible area revealed no significant differences between the complexes. To refine our estimations of the binding energies predicted by docking, we applied MMGBSA (Molecular Mechanics/Generalized Born/Surface Area) on the trajectories.

### 2.3. MMGBSA of the Simulated BG-Pesticide Complexes

MMGBSA is a method to estimate interaction energies from trajectories such as those simulated above. After performing the simulations, these calculations increase the computational load only minimally, as they are carried out in an implicit solvent and ignore solvent entropic contributions. Since these calculations ignore entropy, they result in anomalously big binding free energies that frequently do not correspond to those experimentally determined; however, they are particularly useful for affinity comparisons between different ligands to the same receptor: in this case, the natural ligand palmitate vs. each pesticide. The ΔG values calculated from MMGBSA for all ligands, and the more relevant prediction of ΔΔG values for pesticide vs. palmitate binding, are shown in Figure 5 and Table 3, respectively. The calculations show that six of the twelve pesticides sampled have higher binding energies than palmitate, and four of these are pyrethroids. These pesticides are, in order of increasing affinity, rotenone, **tetramethrin**, fenoxycarb, **permethrin**, **resmethrin** and **cypermethrin** (pyrethroids, in bold). The six remaining pesticides showed lower affinity than palmitate, with the weakest energies found for coumatetralyl and paraquat, which displayed behavior similar to that of the negative control, benzene.

## 3. Discussion

We tested if cow’s milk BLG interacts with the pesticides currently in use across the world. First, by docking, we tested all of the molecules listed as pesticides by the WHO [8] to find “probable ligands”, setting a threshold at −1 standard deviation (−8.18 kcal/mol) of the mean pesticide affinity. Then, we selected representatives of each pesticide class among the “probable ligands”, to sample the stability of their interactions with BLG using molecular dynamics. Last, we evaluated the energetics of the interactions using MMGBSA. Our results suggest that at least three chemical types of pesticides can bind BLG in the calyx: pyrethroids (PY), coumarins (CO) and carbamates (C), which are mostly used as insecticides (PY and C) or rodenticides (CO). Pyrethroids were, by far, the most represented chemical type found in our screen and showed the most stable interactions with and the highest affinities to BLG, representing four out of the six pesticides with higher affinity than palmitate. PY affinity for the BLG calyx likely arises through a mixture of shape complementarity and hydrophobicity, similar to palmitate’s binding mechanism. In contrast, the hydrophilic herbicide paraquat that showed some affinity in docking, was revealed by molecular dynamics analyses to have an unstable interaction with BLG.

The consequences of acute pesticide exposure, either work-related or from household use, are well understood [26], but chronic exposure is far less documented. Exposure to even small doses of the most toxic pesticides, like class Ia coumarin derivatives or organochlorine/organophosphate insecticides, has very obvious acute effects that could limit chronic exposure. Yet, even some very toxic pesticides such as organochlorines and organophosphates are known to bioaccumulate [27]. Interestingly, in the last decade, there has been an overall decline in insecticide use due to a decrease in the most toxic insecticides like organochlorines and organophosphates, which are being gradually replaced by less toxic options, like pyrethroids [5]. PYs are absent from the WHO pesticide class Ia and are distributed in less hazardous classes, including U [8]. PYs are less hazardous in part because of their quick metabolism in humans [28]: their assumed safety has driven PYs to be produced synthetically, and they are currently widely used, despite their higher price [9,29]. In fact, PY use increased during the last decade even though overall insecticide use decreased [5,9].

Cows do not metabolize pyrethroids and excrete them almost intact through feces [30]. PYs are also detectable in milk for over 15 days after a single exposure [8]. Cypermethrin, the ligand that showed the highest affinity to BLG in our screen, is known to cross the blood-brain barrier, causing neurotoxicity and motor deficits in mammalian model organisms [31]. Thus, it is important to evaluate the possibility of chronic pyrethroid exposure due to their bioaccumulation in food sources such as cow’s milk. Since PYs are very hydrophobic (log P ~ 6 to 8), they are likely to associate with fatty acids or hydrophobic protein surfaces in cow’s milk. In our results, cypermethrin and three other PYs tested, showed high affinity for cow’s milk BLG, even beyond that of the natural ligand palmitate. We conclude that pyrethroids are likely to bind BLG and could even compete with its natural ligands, like fatty acids and vitamins.

A typical application of cypermethrin as a cow ectoparasiticide consists of about 0.5 g. Cypermethrin has been reported in raw milk by different authors at concentrations of up to 0.86 μM (0.36 mg/L) [8] and 0.4 μM (0.168 mg/kg) [32], after 24 h of ectoparasiticide application. Even though these concentrations may seem high for a contaminant, they are still lower than palmitate concentrations in milk. Based on data from Månsson [33], 4.2% (*w*/*w*) of bovine milk is fat. Of that fat, 30% (*w*/*w*) is palmitate, which is thus present in milk at about 47 mM. Our theoretical calculations indicate that, mole per mole, cypermethrin would have around 19 kcal more binding energy than palmitate, thus suggesting the possibility of cypermethrin displacing some of the palmitate from BLG. It’s important to evaluate whether BLG-cypermethrin binding can occur in vitro, where the competition between ligands can be measured reliably at relevant concentrations. Furthermore, nothing is known about what happens to cypermethrin in milk during pasteurization and other industrial procedures, like decreaming. However, it is known that ligands such as retinol and folic acid are protected against photodecomposition while interacting with BLG [34,35]. Thus, cypermethrin’s half-life could be extended significantly while bound to BLG.

The ΔG values calculated for palmitate by docking (−7.7 kcal/mol, Figure 2) are not far from those determined experimentally (−8.6 kcal/mol, from isothermal calorimetry [25]). However, the ΔG values calculated through MMGBSA (−33.79 kcal/mol) are almost four times larger than those experimentally determined, since these calculations ignore entropic contributions. As mentioned in the results, MMGBSA’s strength resides in allowing for comparisons of the binding of different ligands on a single receptor (ΔΔG), and not in absolute ΔG value calculations. All of our theoretical calculations of pesticide binding affinities to BLG await confirmation by experimental approaches. Given the uncertainties of Vina and MMGBSA-calculated ΔGs, the Kd calculations for pesticide affinity for BLG would be imprecise. However, our theoretical screening contributes to the narrowing down of pesticide candidates for further experimental work.

Another independent avenue of research that our analysis opens up is the possibility to design biosensors for pyrethroids, coumarins or carbamates, by exploiting BLG’s natural tendency to bind them, and the atomistic knowledge of their binding mechanism. Biosensors are intended to capture a biologically relevant signal and convert it into a detectable signal, such as an electrical one. They involve at least two elements: a biological entity such as a protein, and a transducer, such as an electrochemical transducer. The former detects the signal (performs the chemical interaction), and the latter transforms it into electricity [36]. At least one nanomaterial-based biosensor exists for organophosphorus pesticides, using acetylcholinesterase and gold nanoparticles [37]. BLG is amenable to nanoparticle formation [38], as well as to reengineering to bind selected ligands, such as a dopamine antagonist [39]. BLG has the additional property of being able to bind different ligands with good affinity with a single binding site, which can be exploited in biosensor design. The affinity that we found in BLG for pesticides could potentially be improved by calyx residue modifications.

## 4. Conclusions

Our data support that pesticides such as cypermethrin could be carried in certain food products such as cow’s milk via hydrophobic binding to proteins like BLG. Carrier proteins could increase the real exposure that humans face, even in environments far from where the pesticides are directly used, and beyond their solubility restrictions. Pyrethroids in dairy should be monitored, as their use increases, to replace the more toxic insecticides. Our analysis also opens up the possibility to exploit BLG pesticide binding to design biosensors directed to the types of chemical it binds.

## 5. Methods

In the RCSB database, there are 72 structures for BLG. We identified a subset of 41 structures, defined as those crystallized with ligands (37 structures), or empty but without missing regions in the electron density (4 structures). From those 41 BLG XRD structures obtained from the RCSB website [40], we chose ten with no gaps and with the best stereochemical quality, to evaluate their ability to bind natural ligands, including 6 structures with ligands (1GX8, 1GX9, 1GXA, 2GJ5, 2R56 and 3UEW) and 4 empty (1BSQ, 1BSY, 2BLG and 3BLG). An excel file with a list of all of the structures determined for BLG can be found in Appendix A. Any ligands were computationally removed before docking experiments, and the stereochemistry of these ten structures was checked using Molprobity [41], before and after energy minimization in UCSF Chimera [42]. For pesticide docking, we selected PDB ID 1GXA, since it has the calyx conformation best able to bind diverse hydrophobic ligands with affinities close to those experimentally determined (see results). All pesticide docking results were also replicated on the BLG structure 3UEW and results similar to those for 1GXA were found. 3UEW and 1GXA are the two highest quality BLG structures that have been determined with palmitate.

The 3D structures for pesticides were first built in ChemAxon’s Marvin *Sketch*, and then saved as mol2 files and optimized via quantum mechanics in Gaussian at a DFT/6-31g * level. This last step was performed in order to have optimized charges for use in the parametrization required for the molecular dynamic simulations. The final coordinates were used for docking. Benzene, phenol, phosphate and acetate were used as control molecules during docking. Marvin Sketch was also used to calculate the logP and log D values for all of the ligands. For docking, we employed *Vina* [43], always allowing for ligand flexibility. The docking grid employed was centered in the BLG protein and extended in 30 by 30 by 36 angstroms, covering the entire protein while being centered in the calyx. The exhaustiveness was set to 1000 for all of the runs. The analysis and visualization of the results were performed using UCSF Chimera.

Molecular dynamics were run using AMBER14 [44], specifically using the *pmemd.cuda* module that takes advantage of graphical processing units (GPUs). Typically, 50 ns of simulation time for BLG took 48 h of computer time. BLG was prepared by indicating the two-disulfide bonds present in its native form. The ligands were parameterized with *antechamber* using the result of the optimization step in Gaussian at DFT/6-31g *. The starting coordinates were taken from the docking results. Briefly, simulations were run using the TIP4PEW water model, the AMBER14SB force field for the protein and GAFF for the ligands. Each BLG-ligand complex was solvated in an octahedral water box, with a minimal 10 angstrom distance from the protein surface to the box edge. Once solvated, the water molecules were first relaxed while a 500 kcal/mol constant force restrained the complex. Then, restraints were released, and the solvent and solute were both relaxed. The temperature of the system was slowly raised from 0 to 300K while a 10 kcal/mol constant force restrained the complex. Pressure coupling was introduced while the restraints were released, and the simulation was extended for 200 to 500 ps. After this step, 100 ns production runs were performed. Simulation analysis was carried out using the *cpptraj* version included with AMBER14 [45]. MMGBSA (Molecular Mechanics/Generalized Born/Surface Area) calculations were carried out within AMBER using the *MMPBSA.py* module [46], following the procedure described in its manual. For the calculations, 501 frames out of each trajectory were employed. The analysis of the results was performed with AmberTools15 and visualized with *VMD*. All of the 3D structure figures were prepared using *VMD* [47].

Docking results and representative simulations are available for download at figshare: https://doi.org/10.6084/m9.figshare.11895054 and https://doi.org/10.6084/m9.figshare.11896851.

## Figures and Tables

**Figure 1 ijms-21-01988-f001:**
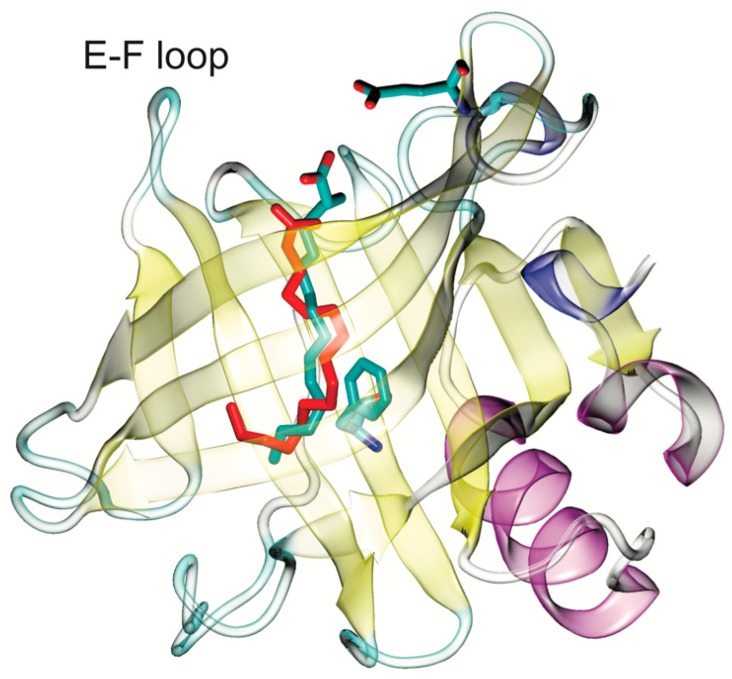
The β-lactoglobulin (BLG) monomer (from PDB-ID 1GXA) with its binding site or “calyx”, formed by eight stranded antiparallel β-sheets and occupied by its natural ligand, palmitate (in green from X-ray diffraction (XRD) and in red from docking). The position of the ligand can be reproduced by docking. Palmitate, and the residues at the bottom (F105) and mouth (E62) of the calyx (used in subsequent calculations), are shown in sticks.

**Figure 2 ijms-21-01988-f002:**
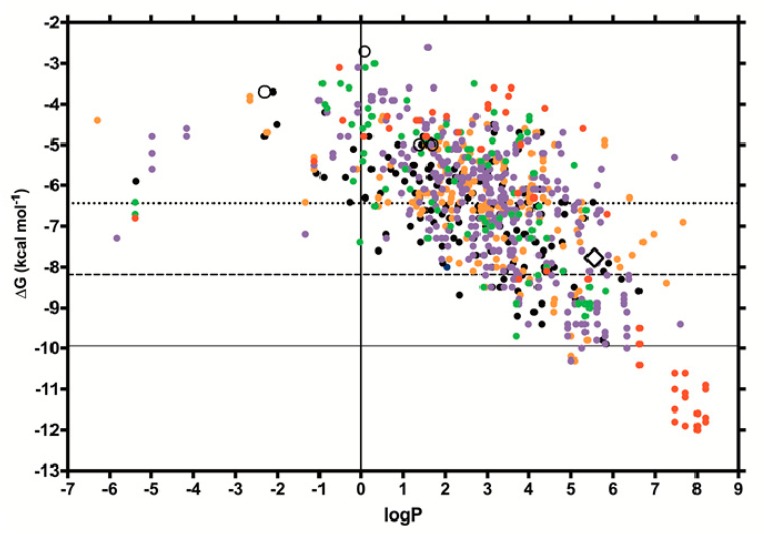
The results of the rigid docking of 889 pesticide molecules to BLG (1GXA). The calculated logP of each molecule is plotted versus the binding energy found through docking. A solid vertical line indicates the logP value where a compound is equally soluble in water and in octanol. The horizontal lines indicate the mean affinity (dotted line at −6.4 kcal/mol), and two standard deviations: −1σ (broken line at 8.1 kcal/mol) and −2σ (solid line at −9.94 kcal/mol). The pesticide class Ia is in red, class Ib in green, class II in purple, class III in orange and class U in black. The palmitate docking result is shown with a diamond, and the negative controls are in empty circles.

**Figure 3 ijms-21-01988-f003:**
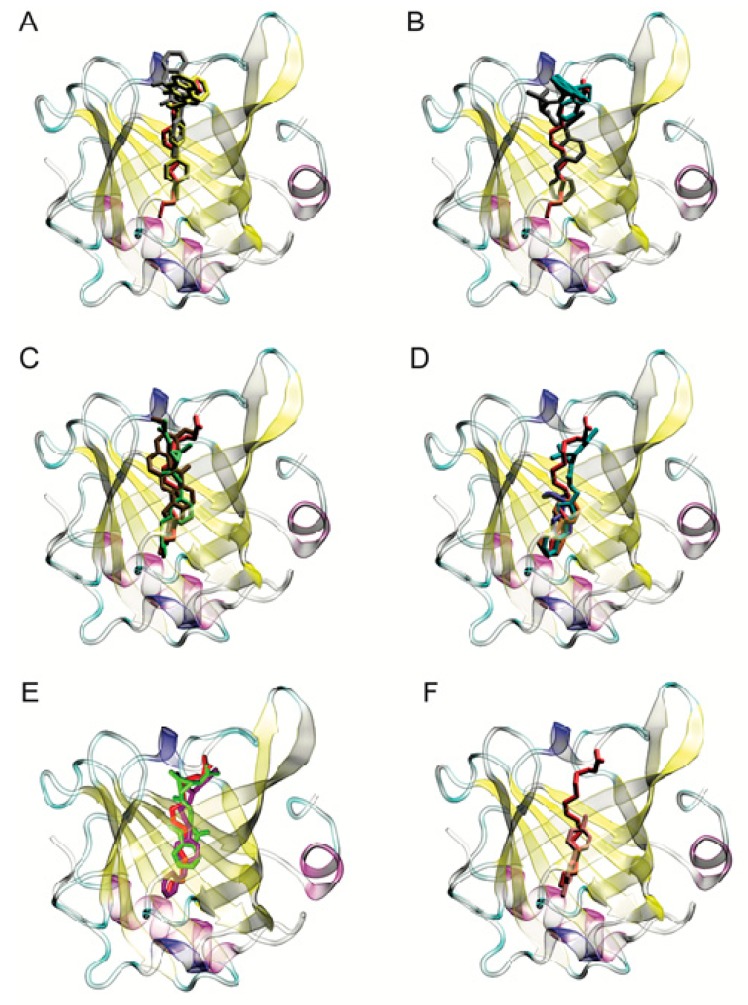
The position of representative pesticides in the BLG calyx, after docking. Eleven representative pesticides identified as “probable ligands” are shown, ordered by class, as in Table 1: (**A**) class Ia, bromadiolone (yellow) and chlorophacinone (gray); (**B**) class Ib, cypermethrin (black) and coumatetralyl (teal); (**C**) class II, permethrin (green) and rotenone (brown); (**D**) class III, biphenyl (orange), cyclohexylbenzene (purple) and resmethrin (teal); (**E**) class U, fenoxycarb (violet) and tetramethrin (lime green). (**F**) shows paraquat, a polar pesticide not selected as a “probable ligand” (pink). Palmitate’s crystallographic position in the calyx is shown for comparison (red in all panels). All ligands are represented by sticks.

**Figure 4 ijms-21-01988-f004:**
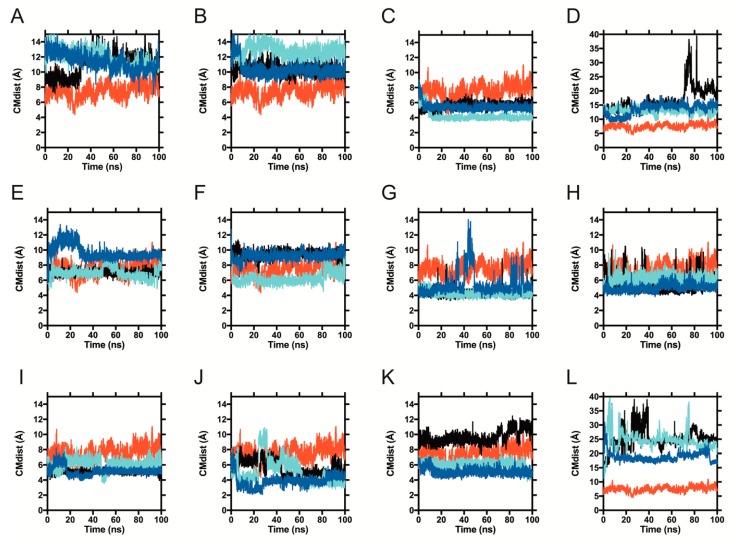
Pesticide position relative to BLG residue F105 during dynamic simulations. The center of mass distance (CMdist) in each of the three molecular dynamics trajectories ran per pesticide is shown throughout time (in light blue, dark blue and black), for representative pesticides: (**A**) Bromadiolone, (**B**) Chlorophacinone, (**C**) Cypermethrin, (**D**) Coumatetralyl, (**E**) Permethrin, (**F**) Rotenone, (**G**) Biphenyl, (**H**) Cyclohexylbenzene, (**I**) Resmethrin, (**J**) Tetramethrin, (**K**) Fenoxycarb and (**L**) Paraquat (same order as in Table 1 and adding paraquat). Palmitate is included in all panels, in red. The scale is the same in all graphs except for (**D**) Coumatetralyl and (**L**) Paraquat, where larger CMdists were observed.

**Figure 5 ijms-21-01988-f005:**
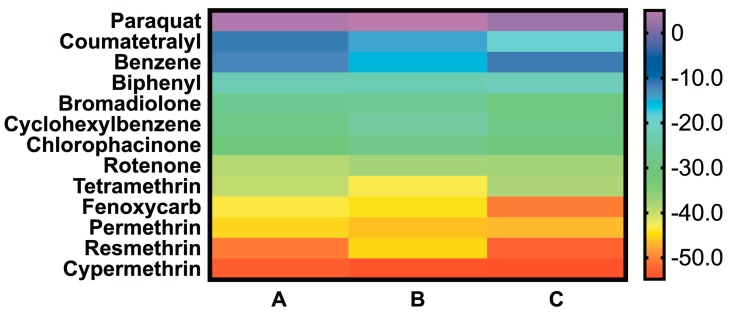
A heatmap of the absolute ΔG calculated from MMGBSA for the 12 simulated pesticides and the negative control, benzene, in triplicate (**A**, **B** and **C**). The highest affinities are shown at the bottom.

**Table 1 ijms-21-01988-t001:** **A** summary of pesticides with binding energies to BLG below −1σ (“probable ligands”).

Class	Representatives, Including Isomers	Unique Representatives, Excluding Isomers	Chemical Types of Unique Representatives, *n* (% of Class)	Simulated Ligands * (Chemical Type)	Chemical Structures of Simulated Ligands
**Ia**	22	6	CO, 3 (50.0) No Type, 3 (50.0)	Bromadiolone (CO) Chlorophacinone	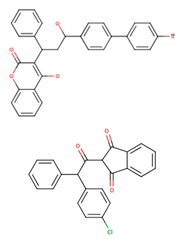
**Ib**	12	5	PY, 4 (80.0) CO, 1 (20.0)	Cypermethrin (PY) Coumatetralyl (CO)	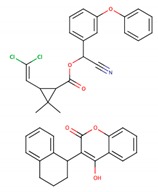
**II**	56	24	No type, 12 (50.0) PY, 11 (45.8) C, 1 (4.2)	Permethrin (PY) Rotenone	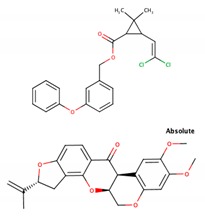
**III**	11	8	No type, 7 (87.5) PY, 1 (12.5)	Biphenyl Cyclohexylbenzene Resmethrin (PY)	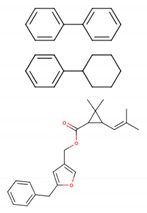
**U**	17	17	No type, 13 (76.5) PY, 3 (17.6) C, 1 (5.9)	Tetramethrin (PY) Fenoxycarb (C)	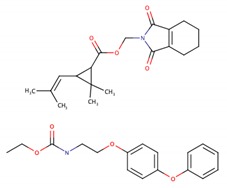
**Total**	118	60	No type, 35 (58.3) PY, 19 (31.7) CO, 4 (6.7) C, 2 (3.3)	11	

PY, pyrethroids; CO, coumarin derivatives; C, carbamates; * shown in subsequent figures and tables.

**Table 2 ijms-21-01988-t002:** Analysis of the CMdist frequency histograms for the simulated complexes of 1GXA-pesticide.

Simulated Ligands	Class	Number of Histogram Peaks	Peak Maxima (Å)
**Bromadiolone**	Ia	2	9.4 and 12.2
**Chlorophacinone**	Ia	2	10.1 and 12.6
**Cypermethrin**	Ib	3	3.9, 5.3 and 7.0
**Coumatetralyl**	Ib	3	12.9, 14.5 and 19.6
**Permethrin**	II	3	6.9, 9.2 and 11.3
**Rotenone**	II	2	6.1 and 9.3
**Biphenyl**	III	2	4.2 and 8.1
**Cyclohexylbenzene**	III	2	4.8 and 5.4
**Resmethrin**	III	2	5.1 and 6.0
**Tetramethrin**	U	4	5.0, 6.2, 9.1 and 10.9
**Fenoxycarb**	U	5	2.7, 3.7, 4.9, 6.2 and 9.1
**Paraquat**	II	3	17.0, 24.3 and 32.0

**Table 3 ijms-21-01988-t003:** The ΔΔG between the pesticides and palmitate from MMGBSA repetitions.

Ligand	ΔΔG ^1^	ΔΔG ^2^	ΔΔG ^3^
**Paraquat**	37.82	38.78	38.78
**Coumatetralyl**	22.22	19.36	19.36
**Benzene**	21.51	18.39	18.39
**Biphenyl**	10.76	10.52	10.52
**Bromadiolone**	6.81	7.18	7.18
**Cyclohexylbenzene**	5.25	8.09	8.09
**Chlorophacinone**	3.23	5.96	5.96
**Rotenone**	−4.91	−3.57	−3.57
**Tetramethrin ^#^**	−5.84	−9.55	−9.55
**Fenoxycarb**	−9.78	−10.48	−10.48
**Permethrin ^#^**	−11.38	−12.65	−12.65
**Resmethrin ^#^**	−16.87	−11.24	−11.24
**Cypermethrin ^#^**	−19.19	−20.46	−20.46

**^#^** Pyrethroids. ΔΔG = ΔGpalmitate − ΔGpesticide. ^1^, ^2^, ^3^ indicate different MD replicates.

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
