# Peer review of "Docking and Molecular Dynamics Predictions of Pesticide Binding to the Calyx of Bovine β-Lactoglobulin"

_ijms, 2020, doi:10.3390/ijms21061988_

Round 1

Reviewer 1 Report

This manuscript (MS) presents an interesting computational study to explore possible binding of pesticides to an important protein such as beta-lactoglobulin present in cow’s milk. The problem is well presented and the way it is addressed in the MS represents a significant example of the applied usefulness of computational analyses. The work is well designed and conducted although the presentation of results leave something to be desired as it is next explained.

Figure 1 could provide a more valuable information if it should be redesigned. Given that the protein is kept fixed in docking calculations and its structure is exactly the same in the four images A-D, it could be much more informative to present a single larger image of BLG with the X-ray and docking geometries of the ligand superposed showing also the side chain of F105 in this same image. Just using sticks with different colors for both ligand geometries and F105 should make much clearer the comparison.

The authors state in subsection 2.1 that pesticides with metal ions were excluded from their study because metal ions are not treated correctly by docking programs. This statement should be qualified. Vina which is the protein-ligand docking procedure used in this MS, is able to cope with metal ions in a ligand since this method uses no charges and consider metals as hydrogen-bond donors. While this is not the better way to treat metals, acceptable results for metal-containing organic compounds are often obtained with Vina as it may be found in the literature.

Figure 2 would be greatly improved if color were also used to distinguish ligand classes. The natural ligands of BLG are hardly identified in this plot.

Similar remarks to those concerning Figure 1 above are also relevant to Figure 3. With images of the protein so small and representations of ligands so large, it is hard to see anything of interest in panels A-F. Again, a superposition of ligands displayed as sticks in different colors on a large unique image of BLG should be much more useful. If, in addition, the chemical formulae of the pesticides selected for rendering this superposition were added to the figure, I believe that the authors would convey a more insightful information.

Figure 4 has also considerable room for improvement. In its current form it is extremely hard to see anything useful. Larger size, better choice of scales in axes, use of color, and an overall better design are clearly needed here as this figure is particularly important in the MS. It is even likely that this information might be better shown in two different figures

Line 192: It must be made clear that MMGBSA estimates protein-ligand binding energies from trajectories obtained in MD simulations. In this regard, the statement in line 194 that MMGBSA calculations are “computationally inexpensive” is misleading: one firstly needs MD trajectories to use the MMGBSA approach. As a matter of fact, the authors claim to have used 501 frames from each trajectory in their MMGBSA calculations.

The results summarized in Figure 5 illustrate a serious flaw of binding energies obtained with either MMPBSA or MMGBSA procedures. Since entropy is ignored, anomalously great binding free energies (exaggerated negative values) are obtained. This fact, implicitly addressed in the Discussion section (lines 245-246), should deserve some comments by the authors.

Since docking calculations move internally the ligands around their rotatable bonds and the values of bond distances and angles are not critical at any stage of the computational study presented, I see no point in performing quantum calculations to optimize geometries of pesticides. A simple energy minimization in UCSF Chimera of the 3D structures initially built with Marvin Sketch would have been enough. Please explain.

In Methods section it is stated that MMGBSA (Generalized Born model for implicit solvent) calculations were performed using the MMPBSA (Poisson-Boltzmann to treat implicit solvent) module of AMBER. Please explain.

Other minor issues:

Line 93: The authors state that 43 X-ray crystal structures of BLG are available in the PDB. However, as of January 25, 2020, a search for “beta lactoglobulin” from “Bos taurus” gave 72 X-ray structures in the PDB. Please clarify.

Line 39:  It is written: “Pesticide sales is increased in the last decade…” yet the reference in support of this statement (ref. 2) dates back to 2013.

Ref. 6 is incomplete.

Line 68: the label “loop EF” is neither explained in the text nor shown in Figure 1.

Line 171: Supplemental Figure 2 is erroneously indicated

No legends to supplementary figures are provided.

Summarizing, in my opinion the MS in its current form is not acceptable for publication in the International Journal of Molecular Sciences due to the concerns raised in my comments above. I therefore recommend the authors to address them in a revised version.

Reviewer 2 Report

The manuscript reports docking and molecular dynamics predictions of pesticide binding to the calyx of ß-lactoglobulin. The data could be considered for publication in International Journal of Molecular Science and could be of interest for the community working in the field, but the manuscript requires revision, comments as follows.

 Page1 Line 2, Title: The title should not contain definition, Modify the title or just delete “a major components of cow’s milk”

Page1 Line 23: The author should give a range of ∆G or simply just one average value (not like ∆Gs below -8.1 kca/ mol)

Page1 Line 30: The author suggests the possibilities for pesticides biosensor based on BLG. The author mentioned that pesticide used in this study dock in the calyx with ∆G of about -8 kcal/mol. This binding energy corresponds to Kd values in mM range which indicate weak binding. The author should elaborate a bit more (in results section) how this will be used for biosensor design.

 Page1: Introduction: The author should include recent references. There are many articles about β-lactoglobuline published in recent years.  

Page2: Line 57-58, 60: The author should double check the abundance of β-lactoglobuline in Whey and milk proteins in the most recent articles. Ungulate milk is not common term- modify or remove.

Page2: The author should make a NEW FIGURE (as Figure 1) about the structure of most important pesticides used in this study that showed tight and stable interactions with β-lactoglobuline. This will be good for readers.

Page3, Line 101: delete “_ “in the end.

Page5 or 6: The author should make a NEW FIGURE to show the interactions between the compounds (pesticides) with amino acids of BLG.

Page5: The resolution of Figure 3 is very bad. Make the structure larger by deleting the gap/space between structures A-E.

Page7: The resolution of the Figure 4 is also not good. Suggest to re-make the figure with visible same font on x- and y-axis.

Page9 Line 229: Cypermethrin was more tightly bound than the natural ligand palmitate. The author should calculate Kd values from binding energies for all those compounds that showed stable interactions and put them in one table. In the discussion section I would suggest to compare these Kd values with experimental Kd values from other hydrophobic compounds.

Supplementary data: I cannot see Figure legend in supplementary data. I would suggest to put all three supple figures in the word/pdf document with legends.

There are also many little things such as spacing of references in text, and units of concertation etc that need to be corrected.

It would also strengthen the manuscript if the authors add a few lines in the discussion or in the conclusion describing HOW this study opens way for designing biosensor based on BLG? i.e if the Kd is in mM range (based on ∆G of approx. -8 kcal/mol) then how can the author correlate these with concentration of pesticides in the milk?

Reviewer 3 Report

The authors of the submitted manuscript describe the use of docking and molecular
dynamics to predict if insecticides and herbicides bind to β-lactoglobulin.

The subject is interesting and regarding the increasing use of pesticides and related compounds, it is useful to have a 'easy' method to check the occurrence in milk.

Fig 1 C and D seems not to be important and can be removed.
The 'mean affinity value of -6.4 kcal/mol" does not say much because the sd seems to be very large.
Fig2: the natural ligands cannot be clearly seen, use of color is recommended.
The use of cLogP without an explanation is not very appropriate.
Resolution of fig3 and fig4 is not good. Same x-axis scale is recommended.
MMGBSA is not written out.
Formatting of numbers and units are inconsistent.
What is the concentration of β-lactoglobulin in milk? Because of the 1 to 1 binding of the
tested molecules, their concentration cannot be higher then that value. If concentrations are found larger than that value, binding in milk fat is more likely. Authors should comment on this issue.
Binding tests with found good binders should be performed to validate the results.

Materials and methods can be a bit more detailed.

The manuscript is interesting, and publication is recommended. The raised issues should be resolved and the text should be carefully checked for errors. The English grammar should be improved.

Reviewer 4 Report

In the manuscript, the authors used docking score and molecular dynamics with MMGBSA to evaluate pesticide binding to beta-lactoglobulin, the most abundant protein in cow milk. The pesticides with lower energetic result were considered as probable binders. Through the computational approaches, the author suggested Pyrethroid in dairy should be monitored.

When a pesticide binds to beta-lactoglobulin, it could be both specific and non-specific binding. The method in the paper only discussed the docking to the orthosteric binding sites. Regardless of the accuracy, a large number of non-specific binding is neglected, which leads to false negative prediction.

The paper is lack of novelty and accuracy. Docking and MD have been extensively used for pose prediction, but the scoring methods are well-known inaccurate. The MMGBSA approach also suffers from missing entropic terms, and low reproducibility due to local minima of conformations in MD. Without experimental validation or experimental data support, the prediction is less persuasive and likely false positive.

Overall, it's not convincing to draw the conclusion that Pyrethroid in dairy should be monitored.

Round 2

Reviewer 1 Report

The authors have addressed satisfactorily all the concers raised in my previous review and the manuscript has been significantly improved.

Author Response

We appreciate the comments from the previous revision and the recognition of our improved manuscript.

Thank you.

Reviewer 4 Report

The authors answered my major concerns of the accuracy of the predictions. As a small and rigid protein, it's a reasonable and also attracting work to the community to rank number of pesticides by using docking and further confirming with MD. The manuscript can be considered for publication with minor revisions.

  1. Line 108 and line 376, 43 and 74 XRD structures have been mentioned. It will be helpful to provide a summary list of the known crystal structures in Supplementary Information with their PDB IDs, X-ray resolutions and name of 2D structures of the ligands if exist.
  2. The lipophilic pocket has been discussed many times in the manuscript. Figure 1 should add a surface plot to show the hydrophobic pocket and the hydrophilic entrance.
  3. In supplementary information figure 1, the plot from 3BLG gave closer energetic prediction to the green reference line of experimental result than 1GXA. The specific reason for why 3BLG is not selected should be given in the manuscript.
  4. The docking energetic result of the 4 negative control molecules should be either marked out in Figure 2, or given in the text.
  5. When analyzing the MD simulation trajectories, it's a common practice to align the protein and then analyze the RMSD change of the ligand heavy atoms to evaluate the stability of the docking pose (e.g. J Comput Aided Mol Des. 2017, 31(2):201; J Comput Aided Mol Des. 2018, 32(1):129). A ligand RMSD plot similar to Figure 4 should be added. It will be informative for the reader for the docking pose stability.
  6. Table 3, the 3 ddG columns should be labeled as MD replicate 1, 2, 3 to clarify the meaning of them.
  7. Suggest the author to upload the most confident docking poses and MD conformations into SI. It will be useful for the community to do following-up study or visualization. 

Author Response

The authors answered my major concerns of the accuracy of the predictions. As a small and rigid protein, it's a reasonable and also attracting work to the community to rank number of pesticides by using docking and further confirming with MD. The manuscript can be considered for publication with minor revisions.

  1. Line 108 and line 376, 43 and 74 XRD structures have been mentioned. It will be helpful to provide a summary list of the known crystal structures in Supplementary Information with their PDB IDs, X-ray resolutions and name of 2D structures of the ligands if exist.

We reviewed the RCDB entries and crafted an xlsx file with the data, that was added as supplementary material. The first tab of the file contains all of the solved structures to date. The second contains those structures showed in Figure S2. We corrected the manuscript to reflect that the structures with ligands that exist for BLG are 37, plus we used 4 without ligands that are complete (without gaps) for a total of 41 initial structures evaluated to choose the 10 best. The total number of Bos taurus BLGs in PDB is 72. We had mistakenly added two structures from horse and reindeer. All the clarifications and corrections are in the text now.

  1. The lipophilic pocket has been discussed many times in the manuscript. Figure 1 should add a surface plot to show the hydrophobic pocket and the hydrophilic entrance.

Added as Figure S1 with Kyte and Doolittle hydrophobicity to represent the inside of the calyx and its mouth. We opted to put this figure in the supplementary material because we consider that the extension of the hydrophilic surface in the calyx mouth is overrepresented by the Kyte and Doolittle plot. We decided not to distract the reader with that discussion. In the main text references that discuss calyx residues and describe the experiments that define its hydrophobicity are pointed out, as in our opinion they are the best evidences of the apolar nature of the calyx.

  1. In supplementary information figure 1, the plot from 3BLG gave closer energetic prediction to the green reference line of experimental result than 1GXA. The specific reason for why 3BLG is not selected should be given in the manuscript.

We disagree with this comment. Results for 1GXA are represented in black symbols and they are located below all other structures (that is, at higher affinities) and overall closer to the green line of experimental results. Only for one of the 10 ligands tested, retinoate, two structures scored with lower DGs than 1GXA:  1GX9 and 2R56. 1GXA was chosen because it produced binding energies closer to the experimental. It was clarified in methods that all docking results have been replicated with the BLG structure 3UEW which produced very similar results to 1GXA. 3UEW was chosen to test because it is a newer structure with good quality.

  1. The docking energetic result of the 4 negative control molecules should be either marked out in Figure 2, or given in the text.

These negative controls have been added to Figure 2, Figure S2, and to the main text.

  1. When analyzing the MD simulation trajectories, it's a common practice to align the protein and then analyze the RMSD change of the ligand heavy atoms to evaluate the stability of the docking pose (e.g. J Comput Aided Mol Des. 2017, 31(2):201; J Comput Aided Mol Des. 2018, 32(1):129). A ligand RMSD plot similar to Figure 4 should be added. It will be informative for the reader for the docking pose stability.

We thank the reviewer for directing us to the mentioned references. It is the opinion of the authors that such an evaluation (the use of ligand RMSD as a measure of docking stability) is worth testing on BLG; however, it should be first employed on BLGs natural ligands. It might be a good way to measure how different length fatty acids fit inside the calyx. Yet, that is not the scope of the present manuscript. As can be seen in different BLG structures with the same ligand (ie. palmitate in 3UEW, 1B0O and 1GXA), the ligand shows significant RMSD when compared. Thus, for BLG it is unclear if a ligand RMSD limit such as 2.0 angstroms would be adequate so that type of analysis it is still unvalidated.

  1. Table 3, the 3 ddG columns should be labeled as MD replicate 1, 2, 3 to clarify the meaning of them.

We have added that information on the top of the table as well as in the table notes.

  1. Suggest the author to upload the most confident docking poses and MD conformations into SI. It will be useful for the community to do following-up study or visualization.

Agreed, we have created two data sets in figshare:

https://doi.org/10.6084/m9.figshare.11895054

https://doi.org/10.6084/m9.figshare.11896851

We appreciate the comments and recognize the improvement they foster on our manuscript.

Thank you.